# Unified Semantic and ID Representation Learning for Deep Recommenders

## Abstract

Effective recommendation is crucial for large-scale online platforms. Traditional recommendation systems primarily rely on ID tokens to uniquely identify items, which can effectively capture specific item relationships but suffer from issues such as redundancy and poor performance in cold-start scenarios. Recent approaches have explored using semantic tokens as an alternative, yet they face challenges, including item duplication and inconsistent performance gains, leaving the potential advantages of semantic tokens inadequately examined. To address these limitations, we propose a Unified Semantic and ID Representation Learning framework that leverages the complementary strengths of both token types. In our framework, ID tokens capture unique item attributes, while semantic tokens represent shared, transferable characteristics. Additionally, we analyze the role of cosine similarity and Euclidean distance in embedding search, revealing that cosine similarity is more effective in decoupling accumulated embeddings, while Euclidean distance excels in distinguishing unique items. Our framework integrates cosine similarity in earlier layers and Euclidean distance in the final layer to optimize representation learning. Experiments on three benchmark datasets show that our method significantly outperforms state-of-the-art baselines, with improvements ranging from 6% to 17% and a reduction in token size by over 80%. These results demonstrate the effectiveness of combining ID and semantic tokenization to enhance the generalization ability of recommender systems.

## 1 Introduction

In large-scale online platforms such as YouTube (Covington et al., 2016), TikTok (Lin et al., 2023), and Amazon (He & McAuley, 2016), effectively recommending items that align with users' preferences while filtering out irrelevant content is crucial. Traditional recommendation systems predominantly rely on ID tokens, wherein each item is uniquely identified by a distinct token (Rendle, 2010; Kang & McAuley, 2018). However, as the number of items expands, this approach becomes increasingly cumbersome due to the redundancy and sheer scale of the token space.

To overcome these limitations, recent research (Rajput et al., 2024; Singh et al., 2023) has explored the use of semantic tokens as an alternative to ID tokens. Nevertheless, existing works face challenges, such as inconsistent performance improvements and issues with item duplication. For instance, TIGER (Rajput et al., 2024) introduces semantic tokens within a deeper and more complex model architecture, making it difficult to isolate the benefits of semantic tokens themselves. Consequently, the

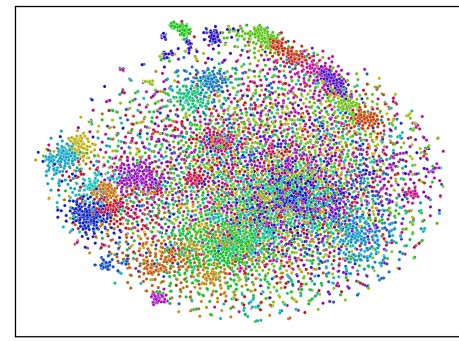

Figure 1: Visualization of ID tokens on Amazon Beauty dataset. Here some ID tokens with the same color share a close embedding space, which means they can be compressed and represented with shared semantic tokens.

true advantages of semantic tokens over ID tokens remain underexplored. Another study (Singh et al., 2023) demonstrates that semantic tokens offer notable improvements primarily in cold-start scenarios, yet both studies report that semantic tokens can map multiple items to the same token, leading to duplication. These open questions invite further investigation into the comparative effectiveness of semantic and ID tokens: *Are semantic tokens inherently superior to ID tokens in recommendation tasks?*

In reality, semantic tokens and ID tokens complement each other. ID tokens have two primary advantages: (1) they can uncover unique, implicit relationships between items, such as the well-known association between beer and diapers, and (2) they facilitate the distinction between different items. However, ID tokens struggle to capture shared attributes across similar items and often suffer from redundancy at scale. This is analogous to whole-word tokenization in Natural Language Processing (NLP)(Bengio et al., 2000; Collobert & Weston, 2008; Mikolov et al., 2013), which tends to fail with unknown or out-of-vocabulary words(Mielke et al., 2021), making ID tokens less effective in cold-start situations. On the other hand, semantic tokens resemble sub-word tokenization in NLP (Mikolov et al., 2012; Wang et al., 2020), where combinations of existing semantic tokens can represent new or unknown items. However, the drawback of semantic tokens lies in their tendency to map multiple, similar items to identical representations, thus failing to distinguish between them (Singh et al., 2023). In summary, while semantic tokens excel in generalizing to unknown items, they are less effective in memorizing unique ones, suggesting that neither approach is universally superior.

To harness the complementary strengths of both token types, we propose a hybrid framework that unifies ID and semantic tokens. As illustrated in Figure 1, our approach begins by visualizing the distribution of ID tokens, revealing that certain items cluster closely together in the embedding space. From this, we hypothesize that only a few dimensions of the ID token space are needed to capture unique item characteristics, while the remaining dimensions can be replaced by semantic tokens to represent shared features. Based on this hypothesis, we introduce a Unified Semantic and ID Representation Learning framework, which incorporates two key components: *unified ID and semantic tokenization* and *unified cosine similarity and Euclidean distance*. First, in unified tokenization, we quantize item content embeddings into a semantic codebook to capture shared characteristics, while assigning each item a low-dimensional ID token to capture unique attributes. Second, in the unified similarity and distance metric, we observe that cosine similarity is effective at disentangling densely clustered embeddings, yet struggles with distinguishing unique items, while Euclidean distance excels at the latter. Consequently, we apply cosine similarity in the earlier layers to decouple dense embeddings and Euclidean distance in the final layer to distinguish unique items. Experimental results on three benchmark datasets demonstrate that our method outperforms existing baselines by 6% to 17%, while reducing token size by over 80%. Ablation studies further validate our hypothesis, showing that many ID tokens are redundant and can be effectively replaced by semantic tokens to enhance generalization.

In summary, the key contributions of this work are as follows:

- We present the first comprehensive investigation into the complementary relationship between semantic and ID tokens in recommendation systems.

- We propose a novel *unified ID and semantic tokenization* framework that captures both unique and shared item characteristics, alongside a *unified similarity and distance* approach that balances embedding decoupling and item distinction.

- Our method achieves significant performance improvements on three benchmark datasets, outperforming baselines by 6% to 17% while reducing token size by over 80%, thereby enhancing the system's generalization capability.

## 2 Preliminary

**Problem Definition**   Suppose there are $m$ items, and each item $i$ is represented by an encoded sentence embedding $\boldsymbol{x}_i$. Let $i_t$ denote user $u$'s $t$-th interacted item. If user $u$ has interacted with a sequence of items $\mathcal{I}_u = (i_1, i_2, \ldots, i_t)$, with corresponding sentence embeddings $\mathcal{X}_u = (\boldsymbol{x}_{i_1}, \boldsymbol{x}_{i_2}, \ldots, \boldsymbol{x}_{i_t})$, the objective of sequential recommendation is to accurately predict the next item that user $u$ will interact with, based on their previous interaction history. Formally, the problem can be defined as follows:

**Input**: A sequence of items $\mathcal{I}_u = (i_1, i_2, \ldots, i_t)$ that user $u$ has interacted with, along with their corresponding sentence embeddings $\mathcal{X}_u = (\boldsymbol{x}_{i_1}, \boldsymbol{x}_{i_2}, \ldots, \boldsymbol{x}_{i_t})$.

**Output**: The estimated probability $\hat{y}_{u,t+1}$ of the next item that user $u$ will interact with at time step $t+1$.

**ID Tokenization**   Traditional recommender systems often rely on ID tokenization to capture the unique characteristics of each item. In this approach, an item embedding matrix $\{\boldsymbol{e}_i\}_{i=1}^m$ is constructed, where each item $i$ is associated with an embedding vector $\boldsymbol{e}_i \in \mathbb{R}^{1 \times D}$, and $D$ represents the ID embedding dimension. The total embedding size for ID tokenization is thus $m \times D$, where $m$ is the number of items. For a user $u$ with an interaction sequence of items $\mathcal{I}_u = (i_1, i_2, \ldots, i_t)$, we can retrieve the corresponding ID embeddings $(\boldsymbol{e}_{i_1}, \boldsymbol{e}_{i_2}, \ldots, \boldsymbol{e}_{i_t})$ through simple lookup operations in the embedding matrix.

**Semantic Tokenization**   To capture the semantic information of items, recent works have leveraged techniques like RQ-VAE (Rajput et al., 2024) to quantize content embeddings. Specifically, semantic tokenization builds $L$ layers of codebook embeddings, where each layer contains a set of embedding vectors $\{\boldsymbol{e}_k^c\}_{k=1}^K$, with $\boldsymbol{e}_k^c \in \mathbb{R}^{1 \times D'}$. Here, $D'$ denotes the semantic embedding dimension, and the total embedding size for semantic tokenization is $L \times K \times D'$. Since $L \times K \ll m$, semantic tokenization can significantly reduce the embedding size by replacing ID-based embeddings with semantically informed ones. As detailed in Algorithm 1 of Appendix A.1, the RQ-VAE model quantizes the input sentence embedding $\boldsymbol{x}_{i_t}$ and returns the corresponding semantic embedding $\boldsymbol{z}_{i_t}$ for each item in user $u$'s interaction history. It is important to note that the stop-gradient operation, denoted as sg, is applied during the quantization process.

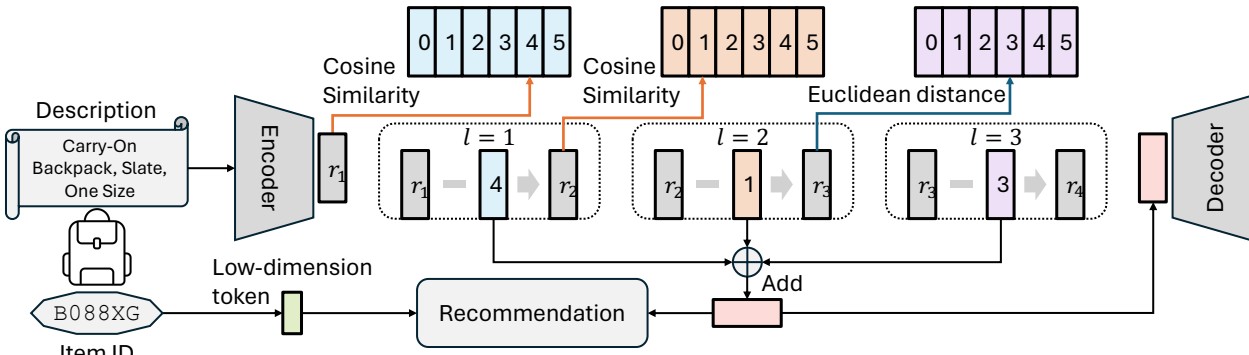

Figure 2: Framework of the unified semantic and ID representation learning. Firstly, the model integrates both semantic tokens, learned through RQ-VAE, and ID tokens for the recommendation task. Secondly, cosine similarity is applied in the first two layers to decouple accumulated embeddings, while Euclidean distance is utilized in the final layer to effectively distinguish unique items. Finally, the overall model is optimized in an end-to-end manner, combining the recommendation loss, RQ-VAE quantization loss, and text reconstruction loss.

## 3   Unified Representation Learning

In this section, as illustrated in Figure 2, we introduce a unified semantic and ID representation learning framework. Our method is designed to fully exploit the complementary strengths of semantic and ID tokens, integrate cosine similarity and Euclidean distance, and jointly optimize both the quantization and recommendation tasks. The key components of the framework are described as follows:

- **Unified Semantic and ID Tokenization**: To balance capturing unique and shared item characteristics, we retain only a small proportion of ID token dimensions to represent the unique attributes of items. Meanwhile, the semantic tokens, learned through RQ-VAE, are employed to capture the shared,

transferable characteristics across items. This hybrid approach reduces redundancy in the ID space while enhancing generalization.

- **Unified Cosine Similarity and Euclidean Distance**: We leverage the strengths of cosine similarity and Euclidean distance in different layers of our model. Specifically, cosine similarity is applied in the earlier layers to effectively decouple accumulated embeddings, while Euclidean distance is employed in the final layer to distinguish unique items. This design maximizes the benefits of both metrics during codebook searching, enhancing the accuracy of item representation.

- **End-to-End Joint Optimization**: Our framework is trained in an end-to-end manner, jointly optimizing three key objectives: (1) the recommendation loss to ensure accurate predictions, (2) the RQ-VAE loss for effective codebook assignment, and (3) the text reconstruction loss to maintain the quality of semantic representation. This joint optimization strategy ensures that all components of the model are fine-tuned for optimal performance in both quantization and recommendation tasks.

## 3.1 Unified Semantic and ID Tokenization

While ID tokenization is effective at capturing unique, item-specific information, it tends to suffer from redundancy and poor generalization, particularly in cold-start scenarios. In contrast, semantic tokenization excels at generalization by capturing shared, transferable features but may introduce item duplication when similar items are mapped to the same token. Therefore, these two approaches are complementary, and combining their strengths can address their respective limitations.

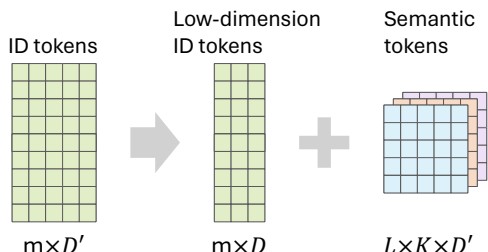

Figure 3: Illustration of unified semantic and ID tokenization. Specifically, we replace ID tokens with low-dimension ID tokens and semantic tokens.

To this end, we propose a unified tokenization strategy that integrates both ID and semantic tokenization. Given that the number of items $m$ can be very large, we reduce the dimensionality of the ID embeddings by setting $D$ smaller than the dimension $D'$ used for semantic embeddings. As shown in Figure 3, our method replaces most dimensions of the ID token with the more generalizable semantic token to reduce redundancy while retaining the ability to capture unique item characteristics. Specifically, for each item $i_t$ in the user's interaction history, we concatenate the semantic embedding $\hat{z}_{i_t}$ and the reduced ID embedding $e_{i_t}$ to form a unified representation, defined as: $s_{i_t} = [\hat{z}_{i_t}, e_{i_t}]$, which results in a sequence of unified embeddings for user $u$, denoted as: $\hat{S}_u = (\hat{s}_{i_1}, \hat{s}_{i_2}, \ldots, \hat{s}_{i_t})$

By combining ID and semantic embeddings, the unified tokenization approach retains the unique characteristics of each item while leveraging the semantic embedding's ability to generalize across similar items. This hybrid representation aims to improve both the efficiency and accuracy of recommendation by reducing redundancy in the ID space and enhancing the model's capacity to generalize to cold-start items.

## 3.2 Unified Distance Function

To enhance the accuracy of codebook selection in our framework, we aim to improve the distance function used for identifying the closest codebook in $k = \arg\min_k \|r_l - e_k^c\|$, as defined in Algorithm 1 of Appendix A.1.

**Statistical Analysis** Our initial analysis, summarized in Table 1, reveals that cosine similarity activates a high percentage of the codebook but struggles to cover unique items effectively. In contrast, Euclidean distance provides high coverage of unique items but activates a much lower percentage of the codebook, with only 5.86% activation in the first layer. The limited activation of Euclidean distance in the early layers may result from its difficulty in decoupling accumulated embeddings, as these embeddings tend to cluster tightly

| Type | Cosine | Euclidean |
|---|---|---|
| First layer | 97.66% | 5.86% |
| Second layer | 98.44% | 100.00% |
| Third layer | 97.66% | 100.00% |
| Total coverage | 70.13% | 92.67% |

Table 1: Comparison of cosine similarity and Euclidean distance in terms of the percentage of activated codebook across three layers and total coverage of unique items. Cosine similarity shows a high percentage of activated codebooks in all layers but lower overall coverage of unique items. In contrast, Euclidean distance exhibits high coverage of unique items, but struggles with a significantly lower percentage of activated codebooks in the first layer.

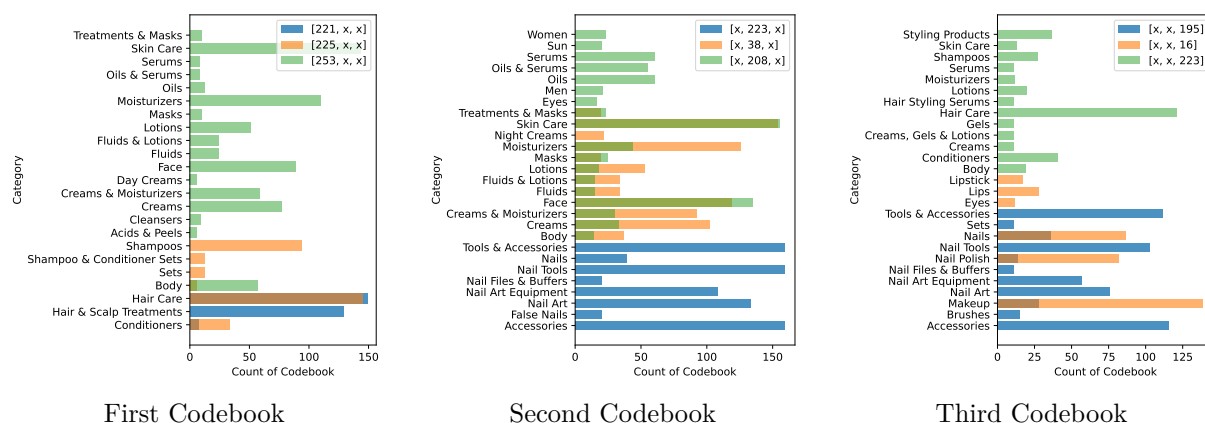

First Codebook      Second Codebook      Third Codebook

Figure 4: Visualization of the codebook selection using cosine similarity across three layers. This figure shows the count of items from various categories assigned to specific token indices, with a focus on the top-3 codebook indices that contain the highest number of items. The distinct distribution of items across different indices suggests that cosine similarity effectively captures category-specific information and helps in distinguishing between categories.

at the beginning. Cosine similarity, on the other hand, excels in decoupling these embeddings, possibly due to its ability to handle orthogonal relationships between embeddings. However, cosine similarity's limited ability to distinguish between distinct embeddings may be attributed to the bounded angular range of 0 to 360°, while Euclidean distance, grounded in the Cartesian coordinate system, provides a more precise measure for distinguishing embeddings based on distance in $\mathbb{R}$.

**Visualized Analysis** To further investigate the performance of cosine similarity and Euclidean distance in codebook selection, we visualized the counts of the top-learned codebooks across different categories using both methods, as shown in Figures 4 and 5, respectively. These visualizations demonstrate that cosine similarity can effectively capture category-specific information across layers, while Euclidean distance struggles to do so in the first layer. Specifically, in the first layer, the codebook entries selected by Euclidean distance appear uniformly distributed across categories, indicating that it fails to differentiate between them.

Based on these observations, we propose the following assumption: *Cosine similarity is more effective at minimizing interference within accumulated embeddings but less capable of distinguishing distinct embeddings, whereas Euclidean distance excels at distinguishing unique embeddings but struggles to decouple accumulated ones.*

**Proposed Method and Experimental Validation** Building on this assumption, we propose a unified approach that combines cosine similarity and Euclidean distance. In the initial layers, cosine similarity is employed to decouple accumulated embeddings, while Euclidean distance is applied in the final layer

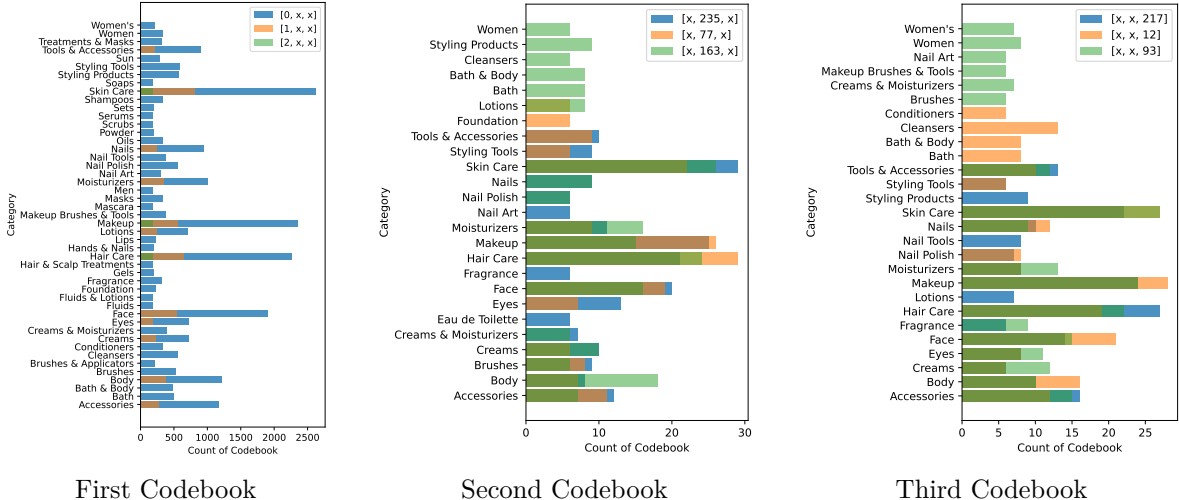

Figure 5: Visualization of the codebook selection using Euclidean distance across three layers. The uniform distribution of items across categories in the first layer indicates that Euclidean distance struggles to effectively capture category-specific information at this stage, making it less capable of distinguishing between categories compared to later layers.

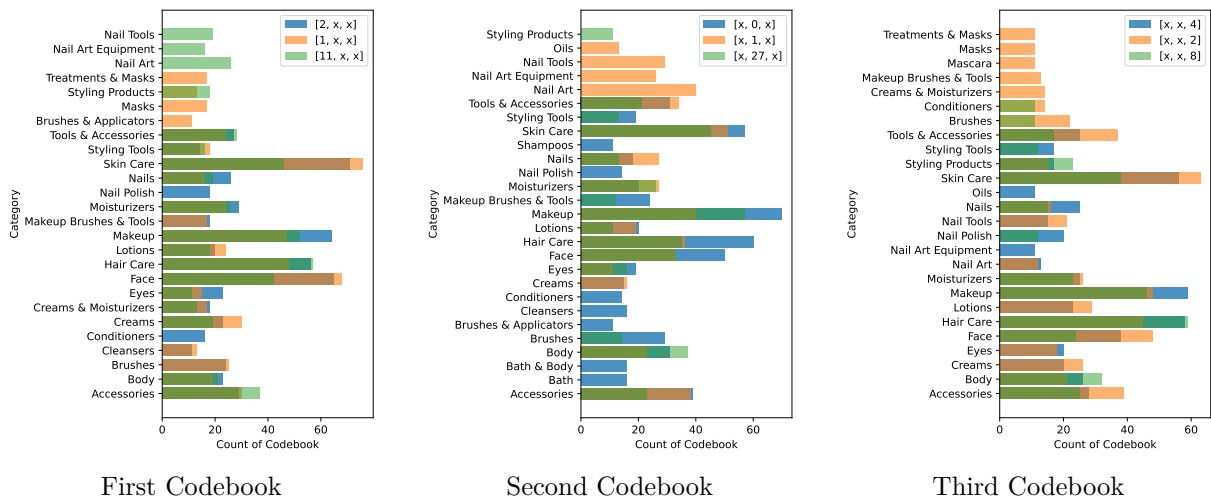

Figure 6: Visualization of codebook selection using the hybrid approach that combines cosine similarity and Euclidean distance. The variation in the counts of items assigned to different codebook tokens across categories demonstrates the effectiveness of this combined method in capturing category-specific information. The integration of both distance measures enhances the ability of Euclidean distance to distinguish between different categories, leading to more accurate item categorization.

to better distinguish unique items. To validate the effectiveness of this hybrid approach, we visualize the codebook selection counts across categories in Figure 6. The results show that the combination of cosine similarity and Euclidean distance successfully captures category-specific information. Moreover, as shown in Table 2, the percentage of activated codebook entries reaches 100%, and the coverage of unique items improves significantly compared to using cosine similarity alone.

**Limitations** Despite the improvements, our proposed method still results in approximately 17% duplicate items, as observed in Table 2. This issue arises when sentence embeddings for certain items are too similar to be distinguished. While this challenge is difficult to completely eliminate, it can be mitigated by assigning

Table 2: Effectiveness of the hybrid approach combining cosine similarity and Euclidean distance. The integration of Euclidean distance into cosine similarity results in a 100% activation of the codebook across layers, while also improving the coverage of unique items. This demonstrates the advantage of leveraging both distance measures for more comprehensive and accurate item representation.

| | | |
|---|---|---|
| Activated codebook | First layer | 100.00% |
| | Second layer | 100.00% |
| | Third layer | 100.00% |
| Coverage of unique items | | 83.27% |

a unique, low-dimensional ID token to each item, helping to further differentiate items with highly similar embeddings.

### 3.3 End-to-end Joint Optimization

After unified tokenization of input item sequence for given user $u$, we then can predict the probability of next item as below.

$$\hat{y}_{u,t} = \Phi(\boldsymbol{s}_{i_1}, \boldsymbol{s}_{i_2}, \cdots \boldsymbol{s}_{i_{t-1}}) \tag{1}$$

where $\Phi$ is the sequential recommendation model to predict the probability $\hat{y}_{u,t}$ of next item. Here $\Phi$ can be any type of sequential recommendation models and we use SASRec (Kang & McAuley, 2018) here. Based on the popular logloss (Kang & McAuley, 2018; Lin et al., 2022), we then can optimize the recommendation model as:

$$\mathcal{L}_{recom} = -\frac{1}{|\mathcal{R}|} \sum_{(u,\mathcal{I}_u)\in\mathcal{R}} (y_{u,t} \log \hat{y}_{u,t} + (1 - y_{u,t}) \log (1 - \hat{y}_{u,t})) + \lambda \|\Theta\|, \tag{2}$$

where $\mathcal{R}$ represents the training set, $\Theta$ denotes the learnable model parameters, and $\lambda$ denotes the regularization hyper-parameter. Finally, we jointly optimize the loss of recommendation, the loss of RQ-VAE, and the loss of reconstruction for text embedding as $\mathcal{L} = \mathcal{L}_{recom} + \mathcal{L}_{rqvae} + \mathcal{L}_{recon}$ (please refer to the algorithm in Appendix A.1).

## 4 Experiments

In our experiments, we evaluate the proposed method on three real-world benchmark datasets, focusing on the following key research questions (RQs):

- **RQ1**: Does the proposed unified representation learning method outperform state-of-the-art sequential recommendation models in terms of prediction accuracy?

- **RQ2**: What is the impact of our unified semantic and ID tokenization method on recommendation performance? Additionally, is the integration of cosine similarity and Euclidean distance effective in improving the final recommendation performance?

- **RQ3**: To what extent can we reduce the dimensionality of ID tokens without compromising performance? Specifically, how does the model's performance vary with different ID token dimensions?

- **RQ4**: What patterns do the semantic and ID tokens learn, and how do these tokens contribute to the overall representation of items?

- **RQ5**: How are the effects of varying the codebook size on the patterns learned by the semantic tokens?

### 4.1 Experimental Setup

**Datasets** We evaluate the recommendation performance on Amazon product review datasets (He & McAuley, 2016), which includes user reviews and item metadata spanning from May 1996 to July 2014. In our task, we focus on three specific categories within this dataset: "Beauty", "Sports and Outdoors" and "Toys and Games." Table 3 presents a summary of the statistics associated with these datasets after applying 5-core filtering, where "Average Len." represents the average length of all users' item sequences. To construct item sequences, we organize users' review histories chronologically by timestamp, ensuring that only users with a minimum of five reviews are retained in our analysis.

Table 3: Data statistics for benchmark datasets after 5-core filtering. Here Sports and Toys are the 'Sports and Outdoors' and 'Toys and Games', respectively, from Amazon review datasets.

| Dataset | # Users | # Items | Average Len. |
|---------|---------|---------|--------------|
| Beauty  | 22,363  | 12,101  | 8.87         |
| Sports  | 35,598  | 18,357  | 8.32         |
| Toys    | 19,412  | 11,924  | 8.63         |

**Evaluation Metrics** We follow the approach used in prior work (Zhou et al., 2020), using Hit Ratio (HIT@k), Normalized Discounted Cumulative Gain (NDCG@k), and Mean Reciprocal Rank (MRR) as evaluation metrics, where $k$ is the number of top ranked items. Consistent with previous studies (Zhou et al., 2020; Lin et al., 2022), given a user behavior sequence, we use the last item for testing, the second-to-last item for validation, and the rest for training. Given the large item set, ranking against all possible items is computationally expensive. Therefore, following a commonly used approach (Kang & McAuley, 2018; Lin et al., 2024), we evaluate the model by sampling 99 negative items along with the ground-truth item. All metrics are calculated based on the ranking of sampled and ground-truth items, and we present the mean scores across users.

**Baselines** To evaluate the pure impact of semantic tokenization, we compare our proposed method against several competitive recommendation baselines, including FM (Rendle, 2010), GRU4Rec (Hidasi et al., 2016), Caser (Tang & Wang, 2018), SASRec (Kang & McAuley, 2018), BERT4Rec (Sun et al., 2019), and HGN (Ma et al., 2019). It is important to note that we do not compare our method with existing work (Rajput et al., 2024) that utilizes a different model architecture with a deeper network when incorporating RQ-VAE. The primary focus here is to examine the effects of semantic tokenization within the context of the same sequential recommendation model to ensure a fair and consistent comparison. Besides, we directly use the results of all baseline from prior work (Zhou et al., 2020) and implement our method based on SASRec under its framework for a fair comparison. Besides, we show the detailed description of these baselines in Appendix A.3.

**Hyper-parameter Settings** We directly use the results of all baseline from prior work (Zhou et al., 2020) and implement our method based on its framework for a fair comparison. Besides, we set some new hyper-parameters of RQ-VAE following prior work (Rajput et al., 2024) with $L = 3$ layers of codebook. We search the codebook size $K$ from 64 to 1024 and select 256 for both Beauty and Toys dataset, while 128 for Sports dataset. Besides, we set the dimension of codebook $D' = 64$ to align with the ID token only method. All other parameters like recommendation model layer and hidden size are set strictly the same as baselines. Additionally, we put more implementation details in Appendix A.2.

### 4.2 Overall Performance

To compare the performance of our method with existing sequential recommenders, as shown in Table 4, we evaluate them in three benchmark datasets under five metrics. From the table, we can have the following observation: **Our method achieves significant improvement.** The improvement of our method towards baselines ranges from 6.07% to 17.87%, which is very significant in sequential recommendation task (Kang & McAuley, 2018; Zhou et al., 2020). Besides, our method improves more on HIT metric than NDCG metric and MRR metric. This may be because semantic embedding is naturally less insensitive at ranking position due to duplicate tokenization, though we have added unique ID embedding.

Table 4: Our method improves baseline significantly by 6% to around 18% on three benchmark datasets.

| Datasets | Metric | FM | GRU4Rec | Caser | SASRec | BERT4Rec | HGN | Ours | Improv. |
|---|---|---|---|---|---|---|---|---|---|
| Beauty | HIT@5 | 0.1461 | 0.3125 | 0.3032 | 0.3741 | 0.3640 | 0.3544 | **0.4201** | 12.30% |
| | NDCG@5 | 0.0934 | 0.2268 | 0.2219 | 0.2848 | 0.2622 | 0.2656 | **0.3079** | 8.11% |
| | HIT@10 | 0.2311 | 0.4106 | 0.3942 | 0.4696 | 0.4739 | 0.4503 | **0.5318** | 12.22% |
| | NDCG@10 | 0.1207 | 0.2584 | 0.2512 | 0.3156 | 0.2975 | 0.2965 | **0.3440** | 9.00% |
| | MRR | 0.1096 | 0.2308 | 0.2263 | 0.2852 | 0.2614 | 0.2669 | **0.3025** | 6.07% |
| Sports | HIT@5 | 0.1603 | 0.3055 | 0.2866 | 0.3466 | 0.3375 | 0.3349 | **0.3849** | 11.05% |
| | NDCG@5 | 0.1048 | 0.2126 | 0.2020 | 0.2497 | 0.2341 | 0.2420 | **0.2717** | 8.81% |
| | HIT@10 | 0.2491 | 0.4299 | 0.4014 | 0.4622 | 0.4722 | 0.4551 | **0.5247** | 11.12% |
| | NDCG@10 | 0.1334 | 0.2527 | 0.2390 | 0.2869 | 0.2775 | 0.2806 | **0.3168** | 10.42% |
| | MRR | 0.1202 | 0.2191 | 0.2100 | 0.2520 | 0.2378 | 0.2469 | **0.2722** | 8.02% |
| Toys | HIT@5 | 0.0978 | 0.2795 | 0.2614 | 0.3682 | 0.3344 | 0.3276 | **0.4340** | 17.87% |
| | NDCG@5 | 0.0614 | 0.1919 | 0.1885 | 0.2820 | 0.2327 | 0.2423 | **0.3141** | 11.38% |
| | HIT@10 | 0.1715 | 0.3896 | 0.3540 | 0.4663 | 0.4493 | 0.4211 | **0.5456** | 17.01% |
| | NDCG@10 | 0.0850 | 0.2274 | 0.2183 | 0.3136 | 0.2698 | 0.2724 | **0.3501** | 11.64% |
| | MRR | 0.0819 | 0.1973 | 0.1967 | 0.2842 | 0.2338 | 0.2454 | **0.3064** | 7.81% |

Table 5: Unified tokenization outperforms ID-only and semantic-only tokenizations with significant reduction of token size. Besides, the semantic tokenization outperforms ID tokenization in position-insensitive metric.

| Dataset | Method | Metric | | | Token Size | | | Token |
|---|---|---|---|---|---|---|---|---|
| | | HIT@10 | NDCG@10 | MRR | ID | Semantic | Total | Reduction |
| Beauty | ID | 0.4654 | 0.3121 | 0.282 | 12,101 × 64 | 0 | 774,464 | \ |
| | Semantic | 0.4956 | 0.2914 | 0.2476 | 0 | 3 × 256 × 64 | 49,152 | 93.65% |
| | Unified | **0.5318** | **0.344** | **0.3025** | 12,101 × 8 | 3 × 256 × 64 | 145,960 | 81.15% |
| Sports | ID | 0.4582 | 0.2826 | 0.2482 | 18,357 × 64 | 0 | 1,174,848 | \ |
| | Semantic | 0.4704 | 0.2554 | 0.2131 | 0 | 3 × 128 × 64 | 24,576 | 97.91% |
| | Unified | **0.5247** | **0.3168** | **0.2722** | 18,357 × 8 | 3 × 128 × 64 | 171,432 | 85.41% |
| Toys | ID | 0.4603 | 0.3092 | 0.2804 | 11,924 × 64 | 0 | 763,136 | \ |
| | Semantic | 0.4644 | 0.2741 | 0.236 | 0 | 3 × 256 × 64 | 49,152 | 93.56% |
| | Unified | **0.5456** | **0.3501** | **0.3064** | 11,924 × 8 | 3 × 256 × 64 | 144,544 | 81.06% |

## 4.3 Ablation Study

To further study the performance of different tokenization methods, we compare our method with the ID tokenization only method and semantic tokenization only method as Table 5. From the table, we can have the following observations:

- **Unified tokenization performs best with significant reduction of token.** In all these three benchmark datasets, our proposed method is significantly superior to solely ID tokenization and semantic tokenization methods. More importantly, compared with the traditional ID tokenization method, our method reduces by at least 80% and even 85% of tokens on Sports dataset. Here we reduce the tokens by replacing 56 dimensions of ID tokens with a small amount of semantic tokens, which supports our previous analysis that most ID tokens are redundant.

- **Semantic tokenization outperforms ID tokenization in position-insensitive metric with significant reduction of token.** In three datasets, it is obvious that the semantic tokenization only method even outperforms ID tokenization only method on HIT metric with less than 10% of tokens. This result also supports our previous analysis that semantic tokenization is effective at generalization and capturing high-level semantic information. However, semantic tokenization only method often performs poor

at NDCG and MRR metrics which are sensitive to position. This is because the position of duplicate tokenized items from semantic tokenization only method are hard to distinguish in ranking.

Table 6: The integration of Euclidean distance into cosine similarity can improve the recommendation performance.

| Method | Beauty | | | Sports | | | Toys | | |
|---|---|---|---|---|---|---|---|---|---|
| | HIT@10 | NDCG@10 | MRR | HIT@10 | NDCG@10 | MRR | HIT@10 | NDCG@10 | MRR |
| Cosine | 0.5212 | 0.3334 | 0.2921 | 0.5129 | 0.3081 | 0.2649 | 0.5252 | 0.3309 | 0.2879 |
| Unified | **0.5318** | **0.3440** | **0.3025** | **0.5247** | **0.3168** | **0.2722** | **0.5456** | **0.3501** | **0.3064** |

Besides, we also compare our method with cosine similarity only method when searching the codebook of RQ-VAE, as shown in Table 6. From the table, we can observe that: **Our unified method outperforms cosine similarity.** The unified method which integrates cosine similarity with Euclidean distance proposed in Section 3.2 outperforms the solely cosine similarity method on three benchmark datasets. This means our unified cosine similarity and Euclidean distance not only can improve the percentage of activated codebook and coverage of unique items, but also can really improve the final recommendation performance.

## 4.4 Hyper-parameter Study

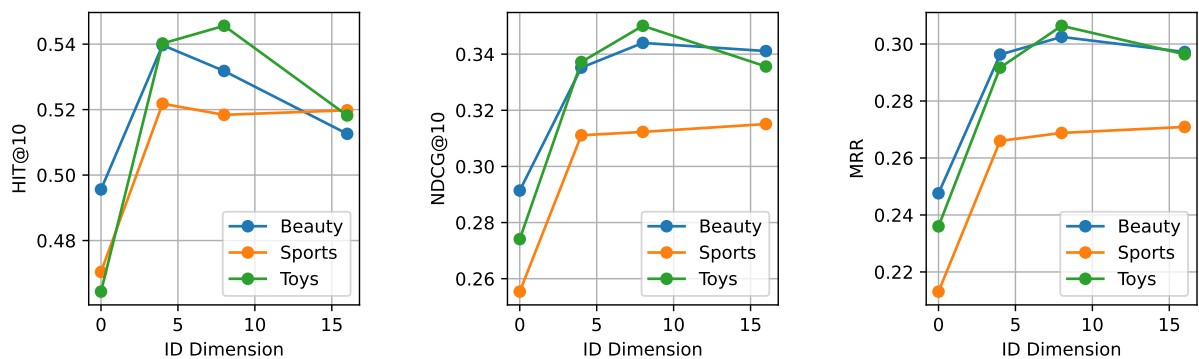

Figure 7: The performance improvement shrinks when scaling up dimension of ID token, which means a small proportion of ID tokens is sufficient for capturing the item's unique characteristic.

To further verify that we only need a small proportion of ID tokens, we further vary the ID dimension from $\{0, 4, 8, 16\}$ and study the performance under three key metrics as Figure 7. From the figure, we discvoer that: **The performance improvement shrinks when scaling up dimension of ID token.** It is obvious that the performance improvement becomes less and less with the growing of ID token dimension, and the performance even drops when dimension is greater than 8. This means a small proportion of ID tokens is sufficient for learning the unique information, and others are indeed redundant and can be saved.

## 4.5 Token Visualization

To study the learned semantic and ID tokens, we further visualize these tokens on Beauty, Sports and Toys datasets using t-SNE, as shown in Figure 8, 9 and 10, respectively. Here we label each semantic token with a unique color and thus these are totally $K$ types of color. In ID tokens, we also label them with $K$ types of color to show the distribution when they are assigned with one of the codebooks. Based on the visualized results, we can discover that:

- **Semantic tokens vary across different layers.** It is obvious that the semantic tokens vary across different layer on all datasets, which means different layers of semantic codebooks can capture various

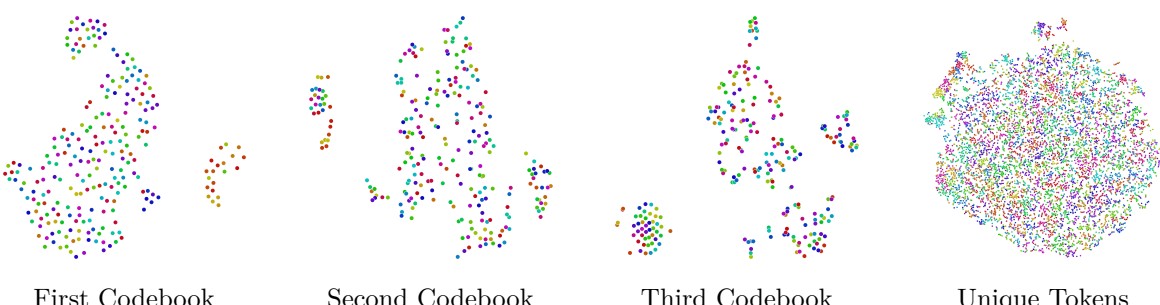

Figure 8: The patterns of codebooks are various across different layers and unique tokens are uniform for different items on Beauty dataset.

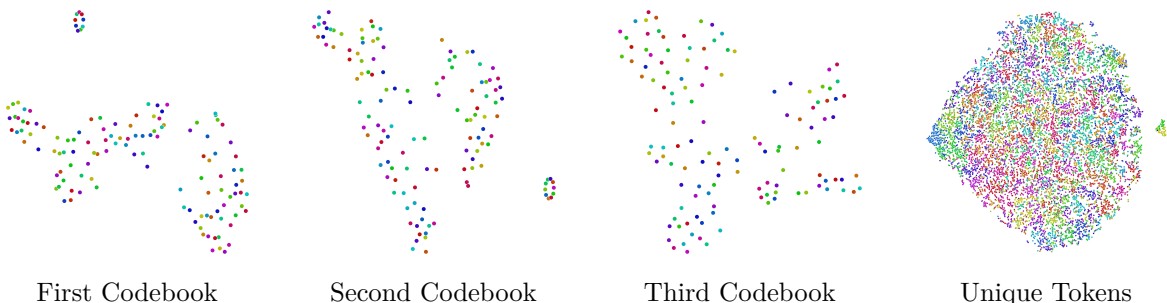

Figure 9: The patterns of codebooks are various across different layers and unique tokens are uniform for different items on Sports dataset.

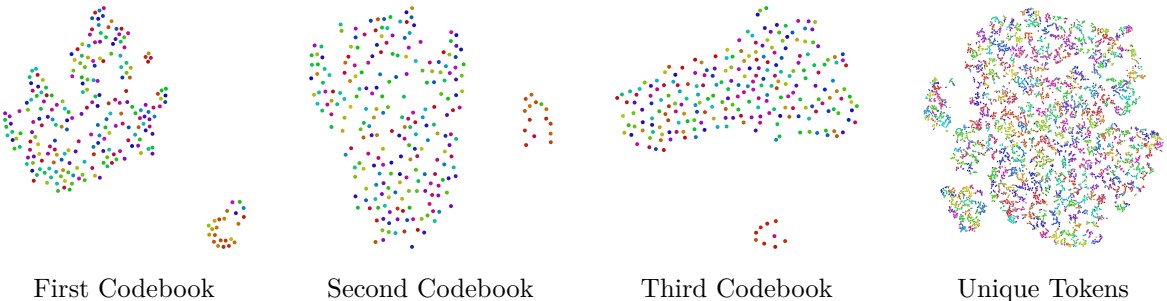

Figure 10: The patterns of codebooks are various across different layers and unique tokens are uniform for different items on Toys dataset.

shared patterns. With the combination of these shared patterns, we can better represent each item's semantic information.

- **ID tokens distribute uniformly.** The unique ID tokens are uniform on all datasets, which means the ID token successfully capture the unique characteristic of each item.

## 4.6 Codebook Size Study

As the first and third codebook in Amazon Sports dataset degenerate in Figure 9, we want to study whether the size of codebook $K$ has significant impact on this degeneration problem. Thus we vary the codebook size $K$ from 64 to 1024 as Table 7, and have the following discovery: **Increasing codebook size does not improve the performance too much.** The performance reaches peak when codebook size is 128, but the performance fluctuates when codebook size grows to 256 and over.

Table 7: Increasing codebook size does not improve the performance too much on Sports dataset.

| Codebook Size | HR@5 | NDCG@5 | HR@10 | NDCG@10 | MRR |
|---|---|---|---|---|---|
| 64 | 0.3792 | 0.2675 | 0.5138 | 0.3109 | 0.2675 |
| 128 | **0.3849** | **0.2717** | **0.5247** | **0.3168** | **0.2722** |
| 256 | 0.3786 | 0.2672 | 0.5184 | 0.3123 | 0.2688 |
| 521 | 0.3842 | 0.2719 | 0.5218 | 0.3163 | 0.2720 |
| 1024 | 0.3809 | 0.2691 | 0.5202 | 0.3140 | 0.2696 |

Besides, we also visualize the token distribution when codebook sizes are 64, 256, 512 and 1024 as Figure 11 to 15 in Appendix A.4. From the figure we can discover that: (1) **The codebooks begin to degenerate and be redundant when codebook size is greater than 256.** The first layer and second layer of codebooks begin to degenerate when codebook size is 256. With the increase of codebook size, the degeneration problem becomes more severe. (2) **The unique tokens are not influenced by codebook size too much.** With the growth of codebook size, the distribution of unqiue tokens almost keep unchange.

## 5 Related Work

**Sequential Recommendation** The use of deep learning in sequential recommendation has evolved into a well-established area of research. GRU4REC (Hidasi et al., 2016) pioneered the application of Gated Recurrent Unit (GRU)-based Recurrent Neural Networks (RNNs) for sequential recommender. Then SAS-Rec (Kang & McAuley, 2018) utilized self-attention mechanisms Vaswani et al. (2017) of Transformer to capture the context relation of whole sequence. Building on the success of masked self-supervised learning in natural language processing, subsequent works such as BERT4Rec (Sun et al., 2019) leveraged self-supervised learning to randomly mask the historical items and improved the robustness. Apart from the popular self-attention and Transformer architecture, researchers have also explored the use of Convolution Neural Networks (CNNs) (Krizhevsky et al., 2012) in sequential recommender (Tang & Wang, 2018). In this paper, we focus on improving the sequential recommendation using semantic tokens.

**Quantized Representation Learning** Vector-quantized learning has grabbed researchers' attention with its discrete latents to reduce the model variance. In recommender systems, VQ-Rec (Hou et al., 2023) proposes a transferable method to quantize item content embedding as item representation. When VQ-Rec utilizes product quantization (Jegou et al., 2010) for the generation of semantic codes, TIGER (Rajput et al., 2024) further leverages RQ-VAE to produce hierarchical semantic IDs as item representation. In parallel to TIGER, another work (Singh et al., 2023) demonstrated that semantic IDs can improve the generalization of recommendation ranking compared with traditional item IDs. Different from existing works aiming to replace item IDs with semantic IDs, we further consider the complementary strengths of them.

## 6 Conclusion

In conclusion, this work provides a comprehensive exploration of the complementary relationship between ID tokens and semantic tokens in recommendation systems, addressing the limitations of using either method in isolation. We introduced a novel framework that unifies ID and semantic tokenization, effectively capturing both unique and shared item characteristics while significantly reducing token redundancy. By leveraging a combination of cosine similarity and Euclidean distance, our approach successfully decouples accumulated embeddings and distinguishes unique items. Experimental results on three benchmark datasets demonstrate that our proposed method consistently outperforms the baselines, achieving notable improvements in performance (6% to 17%) while reducing token size by over 80%. The results also validated our hypothesis that most ID tokens are redundant and can be substituted with semantic tokens to enhance generalization. Our work sets the foundation for a more efficient and effective representation strategy in recommendation systems, combining the strengths of both ID and semantic tokens for improved user experience.

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

# A    Appendix

## A.1    Algorithm for Semantic Tokenization

As shown in Algorithm 1, we present RQ-VAE for semantic tokenization.

---

**Algorithm 1** RQ-VAE for Semantic Tokenization

---

**Input:** Sentence embedding $\mathcal{X}_u = (\boldsymbol{x}_{i_1}, \boldsymbol{x}_{i_2}, \ldots, \boldsymbol{x}_{i_T})$ of user $u$
**Output:** Semantic representation $\hat{\mathcal{Z}}_u = (\hat{\boldsymbol{z}}_{i_1}, \hat{\boldsymbol{z}}_{i_2}, \ldots, \hat{\boldsymbol{z}}_{i_T})$ of user $u$

1: **for** $t = 1 \to T$ in parallel **do**
2:     $\boldsymbol{z}_{i_t} = \textbf{Encoder}(\boldsymbol{x}_{i_t})$ # encode the text embedding
3:     $\boldsymbol{r}_1 = \boldsymbol{z}_{i_t}$, $\hat{\boldsymbol{z}}_{i_t} = 0$
4:     **for** $l = 1 \to L$ **do**
5:         $\{\boldsymbol{e}_k^c\}_{k=1}^K, \boldsymbol{e}_k^c \in \mathbb{R}^{1 \times D'}$ # codebook embedding of each layer
6:         $k = \arg\min_k \|\boldsymbol{r}_l - \boldsymbol{e}_k^c\|$ # search the index of closest codebook
7:         $\boldsymbol{r}_{l+1} = \boldsymbol{r}_l - \boldsymbol{e}_k^c$
8:         $\hat{\boldsymbol{z}}_{i_t} += \boldsymbol{e}_k^c$ # accumulate the quantized embedding
9:         $\mathcal{L}_{\text{rqvae}} += \|\text{sg}[\boldsymbol{r}_l] - \boldsymbol{e}_k^c\|^2 + \beta \|\boldsymbol{r}_l - \text{sg}[\boldsymbol{e}_k^c]\|^2$ # sg means stop gradient
10:     **end for**
11:     $\hat{\boldsymbol{x}}_{i_t} = \textbf{Decoder}(\hat{\boldsymbol{z}}_{i_t})$ # decode the quantized semantic embedding
12:     $\mathcal{L}_{\text{recon}} += \|\boldsymbol{x}_{i_t} - \hat{\boldsymbol{x}}_{i_t}\|^2$ # reconstruction loss
13: **end for**
14: **return** $\hat{\mathcal{Z}}_u$

---

## A.2    Implementation Details

Following TIGER (Rajput et al., 2024), to obtain the semantic tokens, we utilize the pre-trained Sentence-T5 (Ni et al., 2021). Specifically, we construct item's sentence description using its content features, including title, brand, category and price. This constructed sentence is then fed into Sentence-T5, which outputs a 768-dimensional text embedding for each item as the input in our task. Besides, the RQ-VAE model includes a DNN encoder, a residual quantizer, and a DNN decoder. The DNN encoder takes the input text embedding and transforms the dimension to be aligned with codebook embedding. This encoder is activated by ReLU with layer sizes 512, 256, and 128, which ultimately produces a 64-dimensional latent representation. With the 64-dimensional latent representation from encoder, the residual quantizer then performs three levels of residual quantization. At each level, a codebook with size $K$ is used, where each token within the codebook has a dimension of 64. The output semantic token quantized by residual quantizer is then fed into the DNN decoder, which decodes it back to the original text embedding space. Note different from TIGER, we set the dimension of semantic token as 64 for alignment with ID token in our sequential recommendation setting.

As for the implementation of sequential recommendation, we directly use the framework of S³-Rec (Zhou et al., 2020). But as we train the model in an end-to-end manner, we just use the fine-tuning setting and do not use the pre-training setting of their framework. In our setting, we employ the Adam optimizer (Kingma, 2014) with a learning rate of 0.001 and the batch size is set as 256.

## A.3    Baselines

In this section, we provide a brief overview of the baseline models employed for comparison:

- **FM** (Rendle, 2010): The Factorization Machine (FM) model characterizes pairwise interactions among variables through a factorized representation.

- **GRU4Rec** (Hidasi et al., 2016): This model represents the pioneering application of recurrent neural networks (RNNs) for sequential recommendation, specifically utilizing a customized Gated Recurrent Unit (GRU).

- **Caser** (Tang & Wang, 2018): Caser introduces a convolution neural network (CNN) architecture designed to capture high-order Markov Chains. It achieves this through the implementation of both horizontal and vertical convolution operations tailored for sequential recommendation.

- **HGN** (Ma et al., 2019): The Hierarchical Gating Network (HGN) effectively models long-short-term user preference through an innovative gating mechanism.

- **SASRec** (Kang & McAuley, 2018): Self-Attentive Sequential Recommendation (SASRec) employs a causal masked self attention to model user's historical behavior sequence.

- **BERT4Rec** (Sun et al., 2019): This model applies the bi-directional Transformer BERT for enhanced sequential recommender.

### A.4 Token Visualization for Different Codebook Size

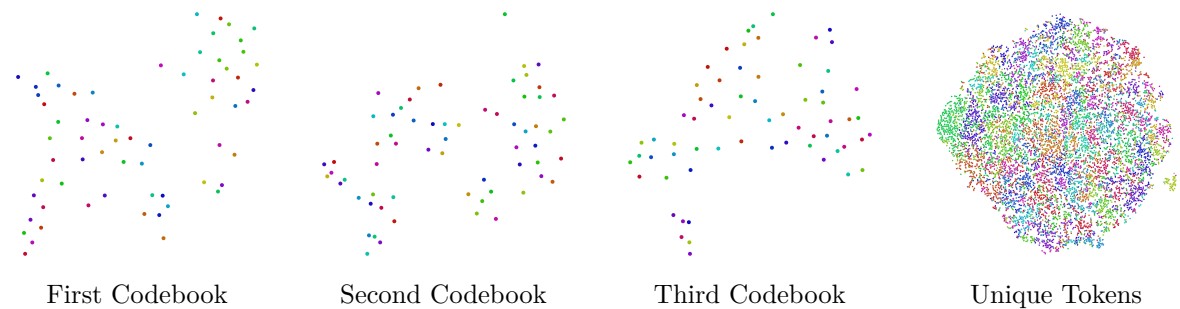

|   First Codebook   |   Second Codebook   |   Third Codebook   |   Unique Tokens   |

Figure 11: The patterns of codebooks are various across different layers but kind of sparse on Sports dataset with codebook size 64.

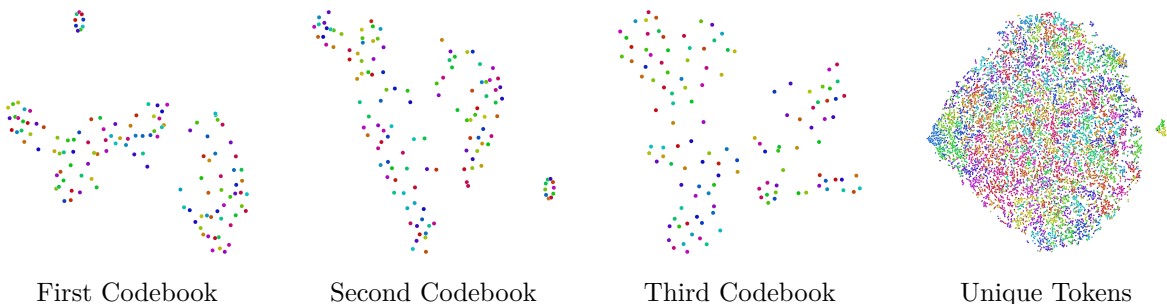

|   First Codebook   |   Second Codebook   |   Third Codebook   |   Unique Tokens   |

Figure 12: The patterns of codebooks are various across different layers on Sports dataset with codebook size 128.

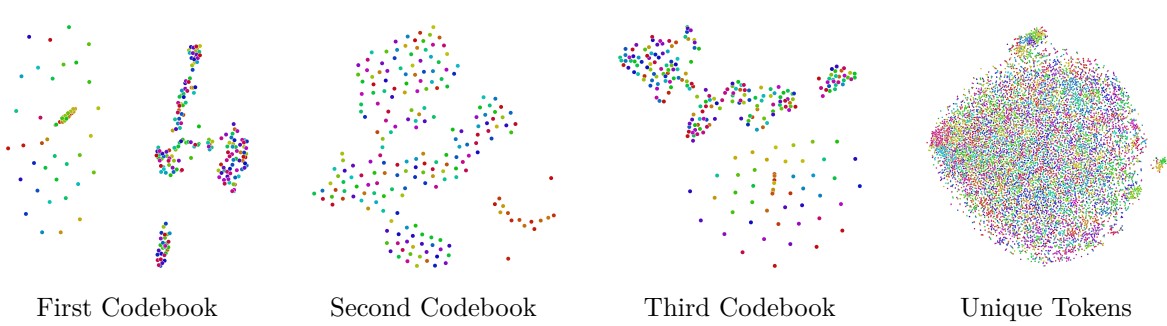

First Codebook   Second Codebook   Third Codebook   Unique Tokens

Figure 13: The first and third codebooks start to degenerate on Sports dataset with codebook size 256.

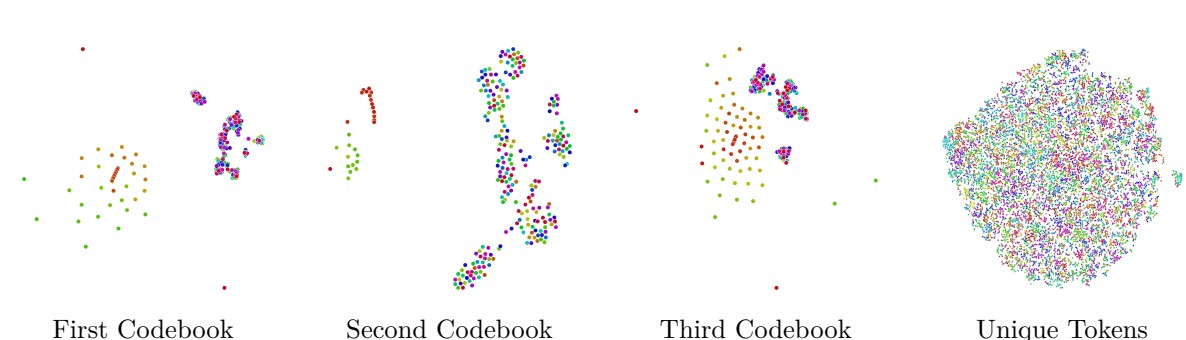

First Codebook   Second Codebook   Third Codebook   Unique Tokens

Figure 14: The first and third codebooks still degenerate on Sports dataset with codebook size 512. And the second codebook also begin to degenerate.

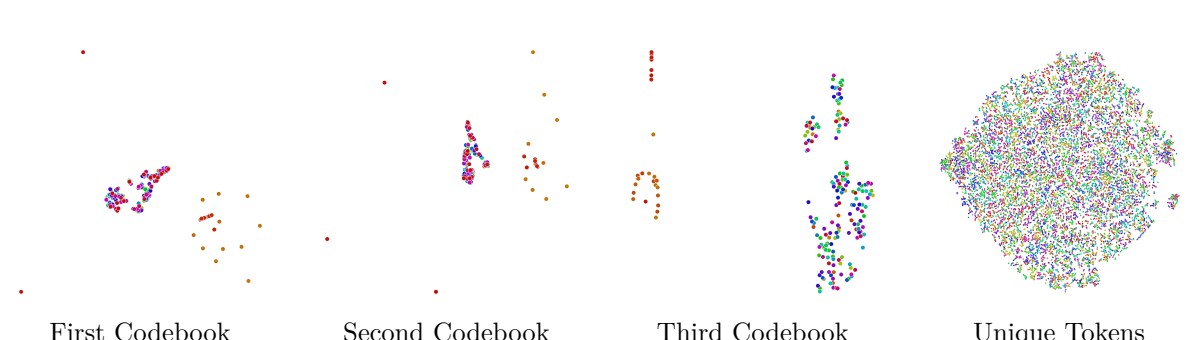

First Codebook   Second Codebook   Third Codebook   Unique Tokens

Figure 15: Almost all codebooks degenerate on Sports dataset with codebook size 1024. In particular, the first and second codebooks degenerate extremely.

