# OpenReview forum: "Unified Semantic and ID Representation Learning for Deep Recommenders"
_TMLR — Under review for TMLR_

### Review · Reviewer_86Ku · 2026-06-20

**Summary Of Contributions:**

The paper studies representation learning for sequential recommendation, with a focus on the complementary roles of \emph{semantic tokens} and \emph{ID tokens}. The central hypothesis is that most dimensions of conventional item ID embeddings are redundant, while a small low-dimensional ID token can preserve item-specific information and semantic tokens can capture shared, transferable item characteristics. To instantiate this idea, the paper proposes a Unified Semantic and ID Representation Learning framework: item content embeddings are quantized by an RQ-VAE into semantic codebooks, and the resulting semantic representation is concatenated with a low-dimensional item ID embedding. The paper further proposes a hybrid codebook-search strategy that uses cosine similarity in earlier RQ-VAE layers and Euclidean distance in the final layer, arguing that the former better decouples accumulated embeddings while the latter better distinguishes unique items. The model is trained with a recommendation loss, an RQ-VAE quantization loss, and a text reconstruction loss.

The main empirical claim is that this unified tokenization framework improves sequential recommendation performance on three Amazon benchmark datasets-Beauty, Sports and Outdoors, and Toys and Games-while reducing the item-token parameter size by more than 80\%. The paper includes ablations comparing ID-only, semantic-only, and unified tokenization; an ablation comparing cosine-only codebook search with the proposed hybrid distance function; a study of ID-token dimensionality; and visualizations of learned semantic codebooks and unique ID tokens.

The paper has several strengths. The problem is timely, since semantic IDs, vector quantization, and generative retrieval are increasingly important in recommender systems. The proposed hybridization of semantic tokens and low-dimensional ID tokens is intuitive and practically relevant. The ablation results suggest that pure semantic tokenization can improve position-insensitive metrics such as HIT, whereas the low-dimensional ID token helps recover ranking-sensitive metrics such as NDCG and MRR. The paper also makes a useful attempt to analyze codebook activation, unique-item coverage, and codebook degeneration.

However, the paper also has substantial weaknesses. The experimental evidence is not yet strong enough to support the broad claims of state-of-the-art performance, improved cold-start generalization, or comprehensive understanding of semantic and ID tokens. The baselines are limited and partly reused from prior work rather than rerun under a fully controlled protocol. The evaluation relies on ranking against 99 sampled negative items rather than full-catalog ranking or a more robust large-negative protocol. The claimed token-size reduction does not appear to account for the RQ-VAE, Sentence-T5 feature extraction, encoder/decoder parameters, training cost, inference cost, or latency. The paper also lacks variance estimates, statistical significance tests, source code, and enough implementation detail to make the empirical results fully reproducible.

**Audience:**

Yes

**Audience Explanation:**

Yes. The topic is relevant to TMLR's audience because it addresses representation learning, vector quantization, recommender-system generalization, and efficient item representation. The interaction between semantic tokens and ID tokens is an important question for modern recommender systems, especially as semantic IDs and generative retrieval become more widely studied. The idea that low-dimensional ID embeddings can complement semantic codebooks is simple, practically motivated, and likely to interest researchers working on sequential recommendation, content-enhanced recommendation, vector-quantized representation learning, and large-scale retrieval.

The findings would be more broadly useful if the authors strengthened the evidence. In particular, a rigorous comparison against recent semantic-ID and vector-quantized recommender methods would clarify whether the improvement comes from the proposed unified tokenization, from the use of content features, from the RQ-VAE objective, or from implementation details. A dedicated cold-start and long-tail analysis would also make the contribution more compelling for recommender-system practitioners.

**Broader Impact Concerns:**

1. Compare against SASRec augmented with unquantized Sentence-T5 item embeddings, randomly initialized low-dimensional ID tokens plus text embeddings, and other simple content-fusion baselines. This would isolate the value of RQ-VAE semantic tokenization from the value of using item text at all.

2. Report training time, inference latency, GPU memory, CPU memory, and throughput as the number of items increases. This would make the efficiency motivation more convincing.

3. The three Amazon categories have relatively short average sequence lengths. Additional datasets with longer histories, different domains, or larger item catalogs would make the conclusions more robust.

4.  The discussion of cosine similarity and Euclidean distance should be sharpened. For example, the claim about the angular range and the claim that Euclidean distance is more precise in Cartesian space are currently informal and not fully convincing in high-dimensional embedding spaces.

5. Figures 4-6 and the t-SNE plots are interesting but difficult to interpret. The authors should add clearer legends, definitions, quantitative summaries, and dataset-specific explanations of what each cluster or color represents.

6. The paper contains several typos and awkward phrases, such as discvoer, unqiue, less insensitive, and fluctuates when codebook size grows to 256 and over. Table 7 lists codebook size 521, which appears to be a typo for 512. Mathematical notation for $\hat{z}_{i_t}$, $s_{i_t}$, $\hat{S}_u$, and the loss terms should be made consistent.

**Claims And Evidence:**

No

**Claims Explanation:**

The paper provides promising evidence for the narrow claim that, under the authors' sampled-evaluation setup on three Amazon categories, concatenating RQ-VAE semantic tokens with low-dimensional ID tokens can improve over the reported SASRec-style baselines. The ablation studies are directionally useful: the unified representation outperforms ID-only and semantic-only variants, and the proposed cosine/Euclidean hybrid codebook search improves over a cosine-only variant. These results are encouraging.

Nevertheless, several central claims are not yet supported by sufficiently accurate, convincing, and clear evidence.

First, the claim of outperforming ``state-of-the-art baselines'' is overstated. The comparison set includes FM, GRU4Rec, Caser, SASRec, BERT4Rec, and HGN, but omits several highly relevant and more recent sequential recommendation or semantic-ID methods, including VQ-Rec, TIGER, semantic-ID ranking approaches, S$^3$-Rec, contrastive sequential recommenders, and stronger content-enhanced baselines. The paper explicitly chooses not to compare against TIGER because TIGER uses a different architecture, but this does not justify claiming state-of-the-art performance. A controlled comparison to architecture-matched semantic-ID variants, as well as comparisons to recent strong sequential recommenders, is needed.

Second, the evaluation protocol is not strong enough for the magnitude of the claims. Ranking the ground-truth item against only 99 sampled negative items can substantially change the absolute and relative values of HIT, NDCG, and MRR. The paper should report full-catalog ranking whenever feasible, or at least evaluate with substantially larger negative sets and multiple negative-sampling seeds. Without this, it is difficult to know whether the reported gains would persist in realistic retrieval/ranking settings.

Third, the paper repeatedly motivates the method through generalization and cold-start behavior, but the experiments do not include a clear cold-start or new-item evaluation. The standard leave-one-out split over user histories does not directly test whether semantic tokens generalize to items unseen during training, to long-tail items, or to domains with sparse behavioral signals. Therefore, the cold-start and generalization claims should either be toned down or supported with dedicated cold-start and long-tail experiments.

Fourth, the reported token-size reduction is not fully convincing as an efficiency claim. The calculation appears to compare item ID embedding parameters with the semantic codebook plus low-dimensional ID embeddings, but it does not clearly include the RQ-VAE encoder/decoder, Sentence-T5 item-text embedding pipeline, additional training objectives, preprocessing cost, inference-time lookup or decoding cost, memory footprint, or latency. If the intended claim is parameter reduction only in the trainable item-token table, this should be stated precisely. If the intended claim is system-level efficiency, the paper needs end-to-end memory, speed, and latency measurements.

Fifth, the paper lacks statistical reliability analysis. No standard deviations, confidence intervals, significance tests, or multi-seed results are reported. Since the gains over some variants are moderate, especially for ranking-sensitive metrics, it is important to show whether the improvements are stable across random initialization, negative samples, and hyperparameter choices.

Sixth, the explanation for the hybrid distance function is underdeveloped. The empirical codebook-activation analysis is interesting, but terms such as ``activated codebook,'' ``coverage of unique items,'' and ``duplicate items'' need precise definitions. The rationale that cosine similarity decouples accumulated embeddings whereas Euclidean distance distinguishes unique items should be supported by more systematic ablations across all possible layer choices, across all datasets, and preferably with a clearer geometric or optimization-based explanation.

Overall, the paper contains a promising idea and useful preliminary evidence, but the current empirical package does not yet support the strongest claims made in the submission.

**Requested Changes:**

1. The paper should compare against recent and directly relevant methods, including VQ-Rec, TIGER or an architecture-controlled TIGER-style variant, semantic-ID ranking methods, S$^3$-Rec, and strong recent sequential recommendation baselines. If some methods are not comparable because of architectural differences, the authors should create controlled variants or substantially weaken the ``state-of-the-art'' claim.

2. The authors should report full-catalog ranking results whenever computationally feasible. If full-catalog ranking is impossible, they should evaluate with substantially larger negative sets, report sensitivity to the number and sampling distribution of negatives, and use multiple negative-sampling seeds. The current 99-negative sampled evaluation is insufficient to support strong ranking claims.

3. The paper should report mean and standard deviation over multiple random seeds, together with significance tests or confidence intervals for the main comparisons. This is especially important for the ablation studies and for metrics where the gains are relatively small.

4. Since the motivation repeatedly emphasizes generalization and cold-start recommendation, the paper should evaluate item cold-start, new-item recommendation, long-tail item performance, or cross-domain transfer. If such experiments are outside the intended scope, the authors should remove or substantially qualify the cold-start and generalization claims.

5. The paper should distinguish between item-token table size, trainable parameter count, total model parameters, preprocessing cost, training cost, inference-time memory, and inference latency. The current ``over 80\% token-size reduction'' claim should not be presented as system-level efficiency unless the RQ-VAE, Sentence-T5 embedding pipeline, and all additional components are included in the accounting.

5. The paper should provide code or a clear reproducibility package. At minimum, it should specify all preprocessing details, maximum sequence length, loss weights, the RQ-VAE commitment coefficient, optimizer settings, training schedule, validation protocol, random seeds, negative-sampling procedure, hardware, runtime, and how semantic codes are updated during end-to-end training.

6. The paper should evaluate Euclidean-only search, cosine-only search, and all layer-wise combinations of cosine similarity and Euclidean distance, not only the proposed configuration. These ablations should be reported on all datasets. The definitions of codebook activation, unique-item coverage, and duplicate-item rate should be made precise.

7. The paper reports that approximately 17\% of items remain duplicated under the proposed hybrid codebook search. The authors should analyze where these duplicates occur, how they affect HIT, NDCG, and MRR, whether they disproportionately affect similar or popular items, and whether the low-dimensional ID token fully resolves the ranking ambiguity.

8. The claim of being the ``first comprehensive investigation'' should be moderated unless the related work and experiments are expanded substantially. The paper should more carefully position itself relative to VQ-Rec, TIGER, semantic IDs for ranking, and other content-enhanced or quantized recommendation methods.

---

### Review · Reviewer_A8HL · 2026-06-25

**Summary Of Contributions:**

This paper studies hybrid item representations for sequential recommendation. The central idea is to combine a low-dimensional learned item-ID embedding, intended to preserve item-specific information, with a semantic embedding derived from content features through RQ-VAE quantization. The paper further proposes using cosine similarity in the early residual-quantization layers and Euclidean distance in the final layer. The resulting representation is used in a SASRec-based sequential recommender and jointly optimized with recommendation, quantization, and reconstruction losses.

The empirical story is clear at a high level. On three Amazon subsets, the proposed unified representation improves over the reported baselines in Table 4, and Table 5 shows that it outperforms both ID-only and semantic-only variants while substantially reducing the reported item embedding-table footprint. Table 6 provides an ablation supporting the mixed cosine/Euclidean design, and Figure 7 suggests that a relatively small ID embedding can suffice under the reported setup.

The main strengths are the intuitive motivation, the relatively lightweight hybrid representation, and the useful ID-only versus semantic-only versus unified ablations. The main weaknesses are that the paper makes broader novelty and generalization claims than the experiments directly support, and that the central baseline comparison is not fully convincing because many baseline results are imported from prior work rather than re-run under the exact same experimental protocol.

**Audience:**

Yes

**Audience Explanation:**

The topic is relevant to TMLR readers interested in recommender systems, representation learning, vector quantization, semantic identifiers, and sequential modeling. The question of how to combine content-derived shared representations with learned item-specific representations is broadly interesting. In particular, the ID-only, semantic-only, and unified comparison addresses a concrete tradeoff between semantic sharing, compression, memorization, and item discrimination.

The paper would be more broadly useful after strengthening its empirical controls and narrowing unsupported claims. Even in its current form, the core design and the reported ablations are likely of interest to researchers working on recommender representations and discrete latent-variable models.

**Broader Impact Concerns:**

N/A. The work uses standard recommendation benchmarks and does not appear to introduce distinctive broader-impact concerns beyond the usual privacy, fairness, and user-manipulation considerations associated with recommender systems.

**Claims And Evidence:**

No

**Claims Explanation:**

Several narrower empirical claims are supported by the presented experiments, but the paper currently overstates what the evidence establishes.

First, the paper's novelty framing is too broad. The claim that this is the “first comprehensive investigation” of the complementary relationship between semantic and ID tokens is difficult to justify based on experiments with one main sequential-recommendation backbone, one semantic-tokenization family, and three Amazon subsets. The work provides a useful empirical study in a particular SASRec-based setting, but “first” and “comprehensive” are substantially stronger claims than the presented evidence supports.

Second, the paper repeatedly attributes benefits to improved “generalization ability” and motivates semantic tokens through cold-start behavior. However, the experiments use standard next-item prediction splits on the same datasets. They do not directly evaluate item cold-start, unseen-item recommendation, temporal distribution shift, cross-domain transfer, or other out-of-distribution settings. The current results support the claim that the proposed hybrid representation improves standard next-item ranking under the reported setup; they do not directly establish improved cold-start or generalization performance.

Third, the methodology around Table 4 requires strengthening. Section 4.1 states that many baseline results are taken directly from prior work, whereas the proposed method is implemented in that framework. This is weaker than comparing all methods under the same codebase, data preprocessing, sequence construction, negative sampling, evaluation protocol, compute budget, and hyperparameter-tuning budget. This issue is particularly important because the Table 4 gains are central to the paper's claims. There is also a concrete comparability concern: the SASRec result in Table 4 does not exactly match the ID-only result in Table 5, despite these appearing to be closely related settings. The discrepancy may be benign, but it should be explained.

Fourth, the attribution of gains is not fully isolated. The unified representation concatenates a 64-dimensional semantic representation with an 8-dimensional ID representation, whereas the ID-only baseline uses a 64-dimensional embedding. Thus, the unified method appears to have a larger total input representation. Without a capacity-matched ID-only control, part of the gain may be explained by additional representation capacity rather than the semantic/ID complementarity alone. In addition, the proposed model uses title, brand, category, and price through Sentence-T5 and RQ-VAE. A content-aware continuous-embedding baseline would help determine whether gains arise from discrete semantic tokenization specifically or simply from access to item content.

On the positive side, Table 5 provides useful evidence that the hybrid representation is stronger than the paper's ID-only and semantic-only variants under the reported setup. Table 6 consistently favors the mixed distance design over cosine-only codebook search. Figure 7 is also consistent with the claim that a small ID embedding can be effective. These are meaningful results, but the paper should narrow its claims to match the scope of this evidence.

Finally, the explanation of cosine similarity versus Euclidean distance in Section 3.2 needs correction. The argument that cosine similarity is limited by an angular range of 0 to 360 degrees is mathematically inaccurate. For standard cosine similarity, the angle between two vectors lies in [0, 180] degrees, and for normalized vectors, Euclidean distance and cosine similarity induce equivalent nearest-neighbor rankings. The observed empirical behavior may still be real, especially if vector norms are informative or residual distributions differ by layer, but the current geometric explanation is not convincing.

**Requested Changes:**

Critical parts:
1. Strengthen the baseline comparison methodology. Re-run at least the strongest relevant baselines, particularly SASRec and preferably BERT4Rec, under the same codebase, preprocessing, data split, negative sampling, evaluation protocol, training budget, and hyperparameter budget as the proposed method. Report mean and standard deviation across multiple random seeds. At minimum, clearly document which aspects of the imported baseline results are exactly matched and explain the discrepancy between the SASRec result in Table 4 and the ID-only result in Table 5.
2. Either add direct generalization experiments or substantially narrow the claims. The current experiments do not test cold-start or out-of-distribution generalization. To retain the current framing, add explicit item cold-start, temporal holdout, long-tail-item, or cross-domain experiments. Otherwise, revise the abstract, introduction, conclusion, and contribution statements to describe improvements in standard sequential recommendation under the evaluated setting rather than improved generalization ability or cold-start performance.

3. Add capacity-matched and content-aware controls. The unified model appears to use a 64-dimensional semantic representation plus an 8-dimensional ID embedding, while the ID-only baseline uses 64 dimensions. Add controls such as a 72-dimensional ID-only baseline and a unified model constrained to the same total dimensionality as ID-only. Also add a content-aware baseline using continuous Sentence-T5 embeddings, with and without the low-dimensional ID embedding, to isolate the contribution of semantic quantization from the contribution of item-content features themselves.

4. Tone down the broad novelty claims. Claims such as “the first comprehensive investigation” should be narrowed. The study is useful, but its empirical scope is limited to one main backbone, one semantic-tokenization approach, and three Amazon subsets. A more defensible formulation would describe the work as an empirical study of complementary semantic and ID representations in a controlled SASRec-based setting.

Strengthen:
1. Correct and clarify the cosine-versus-Euclidean explanation. Replace the current argument based on a supposed 0-to-360-degree limitation of cosine similarity with a mathematically sound explanation, or present the metric choice as an empirical design finding. Useful supporting analysis could include layer-wise residual norm distributions, codebook usage entropy, assignment concentration, inter-category versus intra-category separation, and codebook-collapse diagnostics.

2. Make the evidence in Figures 4–6 more quantitative. The figures are currently interpreted qualitatively as showing category-specific structure. Please define and measure this claim, for example using category-token mutual information, cluster purity, normalized mutual information, entropy, assignment diversity, or category-conditioned codebook utilization. Connect these statistics directly to Tables 1 and 2.

3. Clarify the efficiency claim. The reported 80%+ reduction appears to refer to item embedding-table size, rather than the memory footprint or runtime of the entire recommender system. Please use precise terminology and, if efficiency is a central claim, report offline encoding/quantization cost and online inference/storage implications.

4. Correct reporting inconsistencies. Table 7 lists codebook size “521,” which is almost certainly intended to be 512 based on the stated search space and surrounding discussion. Please check for related numerical, notation, and implementation inconsistencies.

---

### Review · Reviewer_sgeR · 2026-06-26

**Summary Of Contributions:**

This paper studies how to combine semantic tokens and ID tokens for sequential recommendation. The main idea is that ID tokens are good for keeping item-specific information, while semantic tokens are better for sharing information across similar items and reducing token size. The authors propose a unified representation where each item uses RQ-VAE-based semantic tokens together with a small low-dimensional ID embedding. They also propose using cosine similarity in earlier codebook layers and Euclidean distance in the final layer. The method is tested on three Amazon review datasets: Beauty, Sports, and Toys.

The main strength is that the paper tackles a real and useful problem. Pure ID embeddings can be large and do not generalize well, while pure semantic tokens can merge similar items and hurt ranking. The proposed hybrid design is simple and intuitive, and the ablation showing ID-only vs semantic-only vs unified tokenization is useful.

That said, the paper has several important weaknesses.
W1. The novelty is overstated because prior work has already studied semantic IDs, RQ-VAE item tokenization, memorization/generalization tradeoffs, and code collision issues.
W2. The experimental comparison is also not strong enough: the paper mostly compares against older sequential recommenders, but not against several very relevant semantic-ID or learnable-tokenization methods such as TIGER, VQ-Rec, CoST, LETTER, and ETEGRec.
W3. Also, the paper claims cold-start/generalization benefits, but does not actually run a cold-start or unseen-item evaluation.

**Additional Comments:**

I like the general direction of the paper. The method is simple, and the motivation makes sense. My main concern is that the paper currently claims more than it proves. The contribution would be much clearer if the authors framed it as a practical hybrid semantic-ID representation method, instead of making broad claims about state-of-the-art performance and cold-start generalization without enough evidence.

**Audience:**

Yes

**Audience Explanation:**

Yes, I think some TMLR readers would care about this work. Semantic item tokenization, compact item representations, and generative/sequential recommendation are active topics. The idea of combining semantic information with a small amount of ID-based information is practical and easy to understand. The paper may also be useful to people working on recommender systems where item vocabularies are large and pure ID embeddings are expensive.

However, the current version is not strong enough as a complete paper. The findings are interesting, but the evidence needs to be much stronger before the paper can support its current claims. In particular, the paper should be positioned more honestly against recent semantic-ID and learnable-tokenization work.

**Broader Impact Concerns:**

I do not see major ethical concerns that would block the work. However, since this is a recommender-system paper, a short broader impact discussion would still be useful. Recommendation models can affect user exposure, popularity bias, filter bubbles, and fairness across items or creators. If semantic tokens group similar items together, they may also amplify already-common categories or make it harder for niche items to be recommended. The paper should briefly discuss these risks, especially if the method is meant for large-scale online platforms.

**Claims And Evidence:**

No

**Claims Explanation:**

Some claims are supported, but not all of them. The ablation results do support the basic idea that combining semantic tokens with a small ID embedding can work better than using either one alone. The cosine-plus-Euclidean ablation also gives some evidence that the hybrid distance choice helps compared with cosine-only search.

However, the larger claims are not fully supported. The paper claims strong generalization and cold-start benefits, but the experiments use standard Amazon 5-core leave-one-out evaluation. That does not really test new items, cold-start items, or unseen-item generalization. The paper also claims state-of-the-art performance, but the baselines are mostly older sequential recommendation models. Since the paper is about semantic tokenization, it should compare with recent and directly related semantic-ID/tokenization methods, not only FM, GRU4Rec, Caser, SASRec, BERT4Rec, and HGN.

The evaluation setup is also not fully convincing. The paper uses 99 sampled negatives instead of full-catalog ranking. This can make results look better than they would under a harder ranking setup. The authors also do not report standard deviations, multiple seeds, confidence intervals, or significance tests. So it is hard to know how stable the reported gains are.

There are also some unclear parts in the technical story. The explanation for why cosine similarity should be used in early layers and Euclidean distance in the last layer feels somewhat hand-wavy. The paper talks about “decoupling accumulated embeddings,” but this is not clearly defined. Also, the hybrid method has lower unique-item coverage than pure Euclidean distance in the reported analysis, so the codebook story is more mixed than the paper presents.

**Requested Changes:**

Critical changes needed for acceptance:

Add comparisons with directly related semantic-ID/tokenization methods. At minimum, the paper should compare with or carefully discuss TIGER, VQ-Rec, CoST, LETTER, and ETEGRec. The current comparison against mostly older sequential recommenders is not enough for the paper’s claims.

Add a real cold-start or unseen-item experiment. Since the paper repeatedly motivates semantic tokens using cold-start/generalization, it needs an item cold-start split, unseen-item split, long-tail item split, or item-popularity-based analysis.

Use a stronger evaluation setting. The paper should include full-item ranking if possible, or at least evaluate with much larger negative sets in addition to the 99-negative setup.

Rerun baselines under the same setup. Using baseline numbers from prior work is risky because preprocessing, sampling, tuning, and random seeds can differ. The main baselines should be trained and evaluated under the same pipeline.

Report multiple seeds and variance. The paper should include standard deviations or confidence intervals, and ideally significance tests, for the main results.

Add stronger baselines for the actual tokenization design. Since the contribution is about item tokenization, the paper should compare against semantic-only RQ-VAE, ID-only, hybrid ID+semantic alternatives, contrastive tokenization, and recommendation-aware tokenization.

Clarify the cosine/Euclidean distance argument. The current explanation is not very convincing. The authors should define what “decoupling accumulated embeddings” means and explain why the layer-wise choice is expected to help.

Analyze duplicate-item collisions more deeply. The paper admits that around 17% duplicate items remain, but it does not show how much this hurts ranking. The authors should show examples, error cases, and performance broken down by duplicated vs non-duplicated items.

Give a more complete efficiency analysis. The token-size reduction claim should include not only embedding/codebook size, but also RQ-VAE training cost, Sentence-T5 embedding cost, preprocessing, inference time, and memory/storage overhead.

Add ablations on the loss terms. The paper uses recommendation loss, RQ-VAE loss, and reconstruction loss, but does not clearly show what each one contributes. It should also explain whether the losses use equal weights or tuned weights.



Changes that would strengthen the paper:

Test on more diverse datasets, not only three Amazon categories. The current datasets all have short average sequence lengths and come from the same source.

Add analysis by user sequence length and item popularity. This would help show whether the method helps sparse users, long-tail items, or only common cases.

Clean up the notation. The paper switches between predicting (t) and (t+1), and some notation around the loss and token construction is not fully clear.

Make Algorithm 1 match the actual method. The algorithm looks like standard Euclidean RQ-VAE, while the main method uses cosine in earlier layers and Euclidean in the final layer.

Fix typos and writing issues. Examples include “discvoer,” “unqiue,” “unchange,” “performs poor,” and “less insensitive.” Table 7 also has “521,” which seems likely to be “512.”

Use consistent metric names. The paper uses both HIT and HR in different places.

Make the figures easier to read. Figures 4–6 are crowded, and the labels are hard to follow. Some captions also make stronger claims than the figures really prove.

Soften the “state-of-the-art” and “generalization” claims unless stronger evidence is added.